# New Generation Cardiac Contractility Modulation Device—Filling the Gap in Heart Failure Treatment

**DOI:** 10.3390/jcm8050588

**Published:** 2019-04-29

**Authors:** Diana Tint, Roxana Florea, Sorin Micu

**Affiliations:** 1Faculty of Medicine, Transilvania University & ICCO Clinics, Bd. Eroilor nr.29, 500036 Brasov, Romania; 2Sorin Micu, ICCO Clinics, Str. Scolii nr.8, 500059 Brasov, Romania; micu_sorin@yahoo.com; 3Department of Neuroscience, Physiology and Pharmacology, University College London, London WC1E 6BT, UK; 16@ucl.ac.uk

**Keywords:** heart failure, cardiac contractility modulation, cardiac devices, device implantation

## Abstract

(1) Background: Heart failure (HF) is a major cause of morbidity and mortality throughout the world. Despite substantial progress in its prevention and treatment, mortality rates remain high. Device therapy for HF mainly includes cardiac resynchronization therapy (CRT) and the use of an implantable cardioverter-defibrillator (ICD). Recently, however, a new device therapy—cardiac contractility modulation (CCM)—became available. (2) Aim: The purpose of this study is to present a first case-series of patients with different clinical patterns of HF with a reduced ejection fraction (HFrEF), supported with the newest generation of CCM devices. (3) Methods and results: Five patients with a left ventricular ejection fraction (LVEF) ≤ 35% and a New York Heart Association (NYHA) class ≥ III were supported with CCM OPTIMIZER^®^ SMART IPGCCMX10 at our clinic. The patients had a median age of 67 ± 8.03 years (47–80) and were all males—four with ischemic etiology dilated cardiomyopathy. In two cases, CCM was added on top of CRT (non-responders), and, in one patient, CCM was delivered during persistent atrial fibrillation (AF). After 6 months of follow-up, the LVEF increased from 25.4 ± 6.8% to 27 ± 9%, and the six-minute walk distance increased from 310 ± 65.1 m to 466 ± 23.6 m. One patient died 47 days after device implantation. (4) Conclusion: CCM therapy provided with the new model OPTIMIZER^®^ SMART IPG CCMX10 is safe, feasible, and applicable to a wide range of patients with HF.

## 1. Introduction

Heart failure (HF) is a major cause of morbidity and mortality throughout the world [1]. Despite the latest advances in medical and device therapy, mortality remains high, and the vast majority of patients receiving guideline-directed medical therapy (GDMT) remain symptomatic, mainly due to the limitation in medication up-titration.

HF device therapy includes cardiac resynchronization therapy (CRT) for patients with a left ventricular ejection fraction (LVEF) ≤ 35% and a wide QRS (QRS of 130 msec or longer, optimum > 150 msec) and an implantable cardioverter-defibrillator (ICD) for all patients with a LVEF ≤ 35%. These therapies have been validated in solid clinical trials [2,3,4,5] and were included in the recent heart failure guidelines with a class I indication [6]. Unfortunately, not all the patients with advanced HF have a wide QRS complex and, therefore, they may not benefit from CRT therapy. According to an analysis of the Swedish registry, only one-third of patients with HF have a QRS complex wider than 120 msec [7]. Moreover, one-third of patients receiving CRT are non-responders, thus they remain symptomatic, despite the GDMT [8,9,10].

In recent years, a new therapy—cardiac contractility modulation (CCM)—has become available. This therapy delivers high amplitude non-excitatory biphasic electrical signals during the myocardial refractory period. The signal’s voltage can be set to values between a minimum of 4.0 V and a maximum of 7.5 V, according to the patient’s tolerance (the higher values being preferred), and the pulse duration phase can be programmed to one of four possible values between 5.14 msec and 6.60 msec. The stimulation train generally consists of two biphasic pulses having a total duration of 20.5–22.5 msec.

The system modulates the strength of the contraction of the heart muscle by generating non-excitatory impulses. CCM therapy is delivered at regular intervals throughout the day, for a total of 7 to 12 h.

The mechanisms by which this new device improves cardiac contractility are multifactorial. They mainly involve: (1) acute changes in intracellular calcium handling, achieved by an up-regulating process of the L-type calcium channels and an improvement of calcium uptake into the sarcoplasmic reticulum, and (2) chronic changes in the expression and phosphorylation improvement of the key calcium regulatory pathways and in the restoration of the fetal gene expression profile developed during the HF evolution [11,12,13,14].

Initially only dedicated to patients with sinus rhythm (SR) and a narrow QRS, the device was further developed and adapted to be suitable for patients with atrial fibrillation (AF) and non-responders to CRT (patients with a wider QRS).

In contrast to a pacemaker or defibrillator device, the CCM system is designed to modulate the strength of the cardiac muscle contraction rather than its rhythm. CCM therapy is delivered at regular intervals throughout the day. This new therapy is applicable for patients with a New York Heart Association (NYHA) class of II or III, a normal QRS, a LVEF greater than 20%, peak VO2 ≥10 mL/kg/min, and ventricular ectopics or bigeminies of less than 10,000 per day.

## 2. Experimental Section

The present study protocol was reviewed and received ethical clearance by the Ethical Committee of the Transilvania University of Brasov no 01/18/12/2018.

Five patients with HFrEF under appropriate and stable GDMT, with a NYHA class of III or IV, were supported with the latest generation CCM device—OPTIMIZER^®^ SMART IPG CCMX10 (Impulse Dynamics (USA) Inc. Orangeburg, NY, USA). Before implantation, they were evaluated for other potential uncorrected causes of HF (e.g., treatable coronary lesions and frequent ventricular ectopy). The initial baseline evaluation included an assessment of the NYHA functional class and an echocardiographic evaluation of the heart (including cavities measurements, LVEF and left ventricle volumes, mitral regurgitation, and tricuspid regurgitation). The echocardiographic evaluation was acquired with a Vivid 7 machine (GE Healthcare). A six-minute walk test (6MWT) was performed at the baseline and during the 6-month follow-up assessments. The devices already implanted (implantable cardioverter-defibrillators (ICDs) or cardiac-resynchronization and defibrillators CRT-Ds) were interrogated.

The CCM implant procedure was performed under local anesthesia, after the preparation and sterile isolation of the right precordial region of the chest (the right subclavian area). Two active fixation leads (Tendril ST Jude Medical, St Paul, MN, USA) were advanced via the right subclavian vein into the right ventricle (RV) and secured to the right ventricular septum for sensing the ventricular activity and the bipolar delivery of the CCM signals. For each lead, mapping for an ideal position was performed before the lead fixation. RV lead tips were placed along the septal wall at least 2–3 cm apart and at least 3 cm apart from the defibrillation RV lead. The target zone was the septo-parietal trabeculations in the inferior portion of the septal RV outflow tract. Proper placement was appreciated by multiple left and right oblique views. Electrical testing of the leads included the standard testing for pacemaker leads with a higher focus on the sensing function. The lead with the earliest detection was noted. The leads were connected to the implantable pulse generator (IPG) and the device was implanted in a subcutaneous pocket with the recharging coil facing anteriorly. The OPTIMIZER^®^ SMART IPG CCMX10 delivers non-excitatory electrical signals to the right ventricle during the absolute myocardial refractory period. These impulses enhance cardiac strength by triggering physiological processes in the cardiac muscle cells. The CCM device was activated shortly after implantation.

Since all patients had an implanted ICD and the CCM generates a large voltage signal (5 to 7.5 V with a total train duration of 20.5–22.5 msec), testing for a device–device interaction was mandatory. This cross-talking testing required the CCM to be delivered while the ICD was set with active tachycardia detection and enhanced sensitivity.

A chest X-ray was performed after the procedure in order to check the position of the leads and to exclude complications (e.g., pneumothorax or lead dislodgement). Patients were discharged two days after CCM device implantation. Follow-up visits were scheduled at one month after the implant and every three months thereafter. During these visits, the device was interrogated and an echocardiogram was performed along with a 6MWT at the 6-month follow-up visit in order to assess the efficacy of the therapy.

At the time of implantation, all patients were in sinus rhythm, despite having a history of paroxysmal AF. Eventually, one patient developed permanent AF.

Active CCM treatment was programmed to be delivered daily for at least 7 h, in equally spaced out intervals throughout the day, with a voltage between 5 and 7.5 V, according to the patient’s tolerability, and to aim for at least a 90% CCM therapy delivery.

The baseline demographics and characteristics are presented in Table 1, while the technical details related to the device therapy on these patients are shown in Table 2.

## 3. Results

All five patients were male, with a mean age of 67 ± 8 years. In four patients, the CMP etiology was ischemic; furthermore, three of the patients experienced an old myocardial infarction. Most of them were NYHA class III at the time of the intervention, and all of them were in sinus rhythm.

All patients were treated according to the guidelines, and medication was administered up to the maximum tolerated dose without reaching the recommended threshold values. Patient number 3 (P3) did not tolerate any renin—an angiotensin inhibitor—due to very low blood pressure values.

All patients were supported with an ICD, and three of them were also supported with CRT.

The mean LVEF at admission was 25.6% ± 7.2% (20–35%), and the mean 6MWT distance was 310 ± 65.1 m (290–497 m). HF severity was classified as NYHA class III in three patients and NYHA class IV in two patients (see Table 3).

There were no procedure-related complications. No severe or aggravated tricuspid regurgitation was detected.

### 3.1. CCM and Interaction with Other Devices

Figure 1 depicts a patient chest X-ray with a CRT device on the left side and a CCM device on the right side of the thorax.

Figure 2 shows the electrocardiogram ECG aspect of such a patient. One can observe two stimulus artifacts: the first one provided by the CRT device (before the QRS complex) and the second one on the final part of the QRS provided by the CCM system.

Despite the enlarged QRS, the CCM therapy was delivered, and one may notice that the CCM artifact regularly occurs during the QRS complex (the absolute myocardial refractory period).

In our series we had two cases of cross-talking phenomenon. In P4, the RV pacing lead of the CRT device had to be inactivated after CCM implantation, due to cross-talking between the CRT-RV stimulus artifact and the CCM-RV sensing leads. It should be mentioned that we inactivated only the pacing function, leaving the lifesaving therapy (antitachycardia pacing therapy (ATP) and shock) active. The ventricular synchronism was maintained due to the left ventricular LV stimulation. In P2, one of the CCM-RV leads had to be inactivated due to the cross-talking phenomenon with the RV-ICD lead (double counting). Even so, the CCM therapy was effective.

Figure 3 depicts the ECG of a patient with sinus rhythm and a narrow QRS, who was supported with an ICD and CCM, and shows the CCM artifact which occurred during every QRS complex.

Figure 4, on the other hand, shows the ECG in a patient with persistent atrial fibrillation (AF) and a narrow QRS, who was supported with an ICD and CCM. One can note that the CCM artifact is not present on top of every QRS complex. At the time of the device interrogation, we documented CCM activity in only 74% of the CCM activation time. Despite this, the patient experienced symptom alleviation and an increased exercise tolerance. His CCM device was also re-programmed for a total of 10 h/day for therapy delivery.

### 3.2. Outcome

Table 3 summarizes the patients’ outcome after the 6-month follow-up. Overall, the LVEF increased from 25.4 ± 6.8% to 27 ± 9.1% and the 6MWT from 310 ± 65.1 m to 466 ± 23.6 m. All patients (except P3, who died) reached the functional NYHA’s class II.

P3 did not respond to therapy and died 47 days after implantation. In our opinion, this event was not related to the CCM implant procedure. The patient had the most severe outcome with multiple re-admissions, hyponatremia, and renal kidney failure aggravation; therefore, he was probably too ill to benefit from any therapy except heart transplantation. The four remaining patients showed a clinical improvement after CCM therapy was provided.

P5 had one re-admission due to an attempt at cardioversion, which was aborted due to persistent left atrial thrombosis. However, despite the persistence of atrial fibrillation, his NYHA class and his effort tolerance improved. His CCM device was eventually re-programmed to deliver therapy for 10 h/day.

## 4. Discussion

A significant proportion of patients with HF remain symptomatic despite GDMT, and only one-third may benefit from CRT therapy [7,8,9,10]. CCM therapy was developed for patients with mid-range and reduced LVEF and was, for the first time, included in the European Society of Cardiology’s (ESC) 2016 guidelines for the diagnosis and treatment of acute and chronic heart failure, with an indication for selected patients [6]. The safety and efficiency of the CCM therapy was evaluated in a few trials, which encompassed a relatively small number of patients [15,16,17]. These trials included patients with HF and a mid-range LVEF of 25–45%, a QRS duration of less than 130 msec, and who were still in the NYHA’s class III, despite GDMT.

After confirmation of the efficacy and safety of the CCM therapy in the FIX-5 trial [18], the beneficial effects were once again highlighted in a recent FIX-5 sub-analysis, comprising patients with a moderately reduced LVEF. The implantation of CCM in 74 patients with this condition was safe, improved exercise tolerance and quality of life, and decreased HF hospitalization compared to patients without CCM [19].

Despite the fact that CRT became standard therapy for patients with a LVEF ≤35% and an enlarged QRS, only one-third of HF patients were proven to meet those criteria and, furthermore, some of them were non-responders to therapy. An older version of the CCM device—OPTIMIZER III (a three lead model (one atrial and two ventricular leads))—was evaluated in 16 non-responder patients to CRT, showing that the association of these devices seems feasible, but with some calculated risks, which were mainly related to various complications (e.g., lead dislodgement and arrhythmias) [20].

In a more recent trial, Kuschyck and colleagues evaluated the effect of CCM therapy on the same category of patients, using an older version of the device—OPTIMIZER^TM^. This was a multi-centric trial, which included 17 patients in whom the functional parameters were evaluated after 6 months following the CCM therapy. Exercise tolerance increased, as well as the peak VO2, the LVEF trended upwards, and quality of life (assessed with the Minnesota Living with Heart Failure Questionnaire) improved [21]. Moreover, the study showed that CCM is safe and efficacious in CRT non-responders.

The OPTIMIZER IVs was initially contraindicated for patients with permanent or long-standing persistent atrial fibrillation or flutter, as the device was programmed to 100% VVI pacing.

In contrast with that version, OPTIMIZER^®^ SMART IPG CCMX10 is a new product version for the OPTIMIZER Mini platform. CCM therapy delivered by the OPTIMIZER^®^ SMART is the same as the CCM therapy delivered by the previous OPTIMIZER system. The new system, however, has a two-lead configuration, which allows the atrial sensing lead to be optional. This configuration reduces lead-related complications (e.g., dislodgement, perforation, myocardial damage, mechanical obstruction of the superior caval vein, fracture, and infection), it shortens the procedure, and has a clinical benefit.

Compared to older versions, the new OPTIMIZER^®^ SMART model allows CCM therapy delivery on patients with permanent atrial fibrillation, which formerly was considered a contraindication for the previous generation of OPTIMIZER^®^ device.

In this respect, the atrio-ventricular sequence algorithms can be turned off, and the system enforces stricter parameter ranges for the right ventricle local sensing (RV-LS) velocity filter algorithm to ensure stringent discrimination between the ventricular beats originating in the atrio-ventricular AV node from those that result from the activation of ectopic foci. In order to ascertain the adequate timing of the CCM delivery even in patients with an irregular heart rhythm, some safeguards were implemented in the programming [22]. We demonstrated the beneficial effect of the OPTIMIZER^®^ SMART on a patient with HFrEF and atrial fibrillation.

The majority of patients treated with CCM have HFrEF with an LVEF ≤ 35%, thus most of them have also been supported with an ICD. The devices have to be implanted in two separate procedures, as currently there is no device which combines both therapies. Implanting a CCM on top of an ICD requires a well-established algorithm, meant to minimize the detrimental cross-talk between the two devices. Moreover, in OPTIMIZER^®^ SMART, a ventricular rate limit of 90–110 b/min was imposed as an additional precaution for the delivery of CCM to inhibit the delivery of therapy during possible episodes of slow ventricular tachycardia VT.

In our series, recording the cross-talk between the devices in one patient led to the inactivation of one of the CCM leads, in order to avoid double counting the ventricle signals. Despite this, the CCM therapy seemed to be efficacious even when delivered through a single RV lead. In another patient, the CRT therapy had to be provided only by LV stimulation, leaving the lifesaving therapy (ATP and shock) active.

While the clinical trials mainly included patients with HF and a LVEF between 25% and 45%, we reported the outcome of three patients having a LVEF less than 25%. Two of them had a good clinical outcome, with an improvement in their exercise capacity, while one patient died. Further trials are needed to identify the lowest LVEF below which therapy is no longer beneficial.

## 5. Conclusions

CCM therapy provided with the new OPTIMIZER^®^ SMART IPG CCMX10 model is safe, feasible, and applicable to a wide range of patients with HF. Its interaction with other devices (CRT-D and ICD) requires additional tests, follow-up assessments, and sometimes a re-programming of the device. CCM therapy using the OPTIMIZER^®^ SMART IPG CCMX10 in HF patients with permanent AF may improve clinical symptoms and exercise capacity, but a focus on the total time of delivered CCM therapy is needed.

## 6. Limitations

This was a series analysis involving only a limited number of patients, with a relatively short follow-up, without a randomized control group. It should be noted, however, that, even in previously published trials and studies, the number of patients enrolled was quite small and much lower compared to studies focusing on CRT and/or ICD therapy. Further trials are needed to determine the magnitude of the effect of CCM provided by the new OPTIMIZER^®^ SMART IPG CCMX10 on HF in patients with atrial fibrillation or who fail CRT.

## Figures and Tables

**Figure 1 jcm-08-00588-f001:**
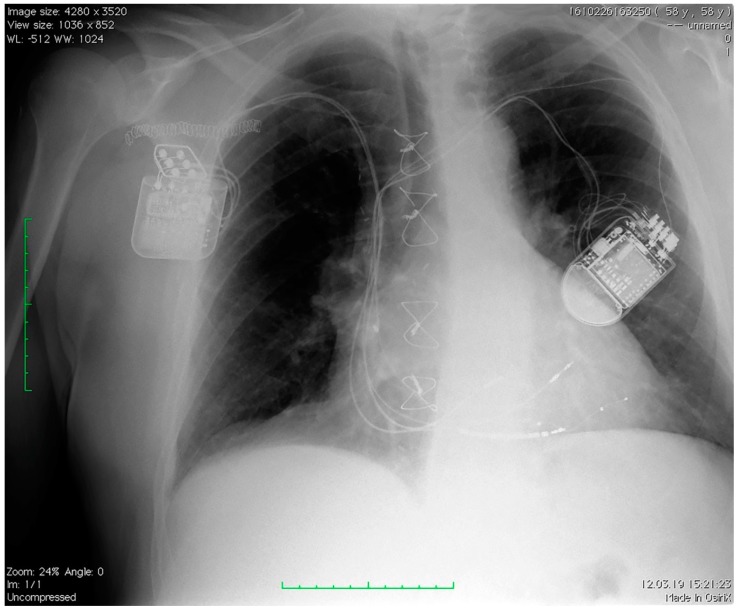
Chest X-ray of a patient with a CRT device on the left side and a CCM device on the right side of the thorax. CRT: cardiac resynchronization therapy; CCM: cardiac contractility modulation.

**Figure 2 jcm-08-00588-f002:**
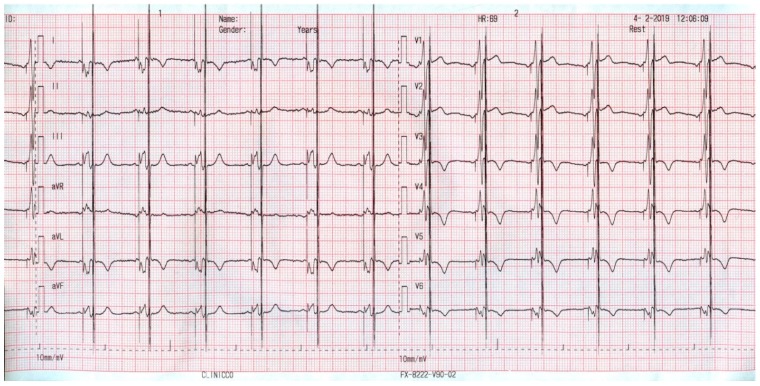
ECG aspect of a patient supported with a cardiac resynchronization therapy and defibrillator (CRT-D) and CCM.

**Figure 3 jcm-08-00588-f003:**
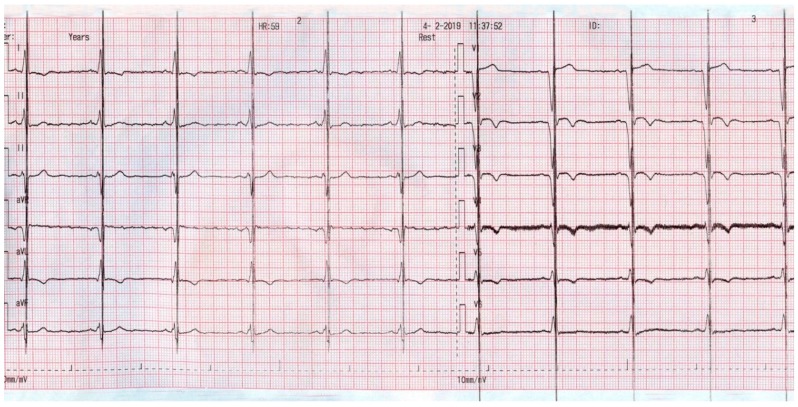
ECG aspect of a patient with sinus rhythm supported with an ICD and CCM.

**Figure 4 jcm-08-00588-f004:**
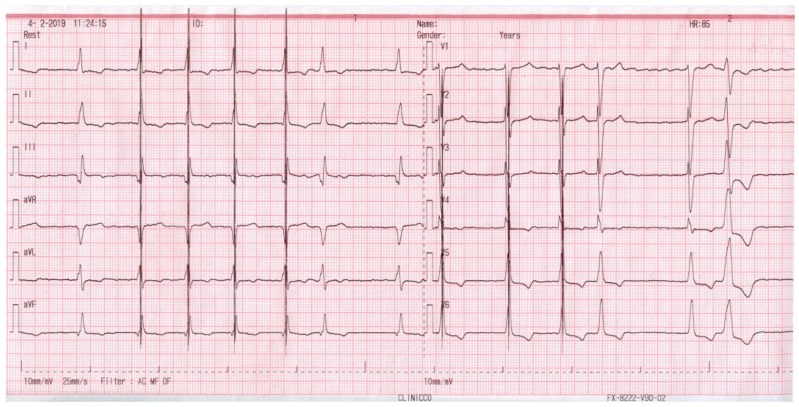
ECG aspect of a patient with atrial fibrillation supported with an ICD and CCM.

**Table 1 jcm-08-00588-t001:** Baseline demographics and characteristics.

	Patient 1	Patient 2	Patient 3	Patient 4	Patient 5
**Age (years)**	69	67	54	69	76
**Etiology of CMP**	ischemic	ischemic	non-ischemic	ischemic	ischemic
**Prior MI**	yes	yes	no	no	yes
**Prior CABG**	no	no	no	yes	no
**Hypertension**	no	no	no	yes	yes
**Diabetes mellitus**	yes	no	no	yes	no
**Creatinine clearance (mL/min)**	78.9	60.3	60.6	88.9	44.1
**QRS width (msec)**	120	130	160	130	120
**Baseline medication**					
**Beta blockers**	yes	yes	yes	yes	yes
**ACE inhibitor/ARB**	yes	yes	no	no	no
**Sacubitril valsartan**	no	no	no	yes	yes
**Aldosterone inhibitor**	yes	yes	yes	yes	yes
**Diuretic**	yes	yes	yes	yes	yes
**Digoxin**	no	no	yes	no	no
**Anticoagulant**	yes	yes	yes	yes	yes
**Statin**	yes	no	no	yes	yes

ACE: angiotensin converting enzyme; ARB: angiotensin II receptor blocker; CABG: coronary artery bypass graft; MI: myocardial infarction; CMP: cardiomyopathy.

**Table 2 jcm-08-00588-t002:** Device therapy—Technical data.

	Patient 1	Patient 2	Patient 3	Patient 4	Patient 5
CRT/ICD device	MAXIMO IIVRD284VRC, MEDTRONIC	MAXIMO II DR, MEDTRONIC	LUMAX 340 HF-T, BIOTRONIK	IFORIA 3 HF-T, BIOTRONIK	IFORIA 3 DR-T, BIOTRONIK
CCM stimulation (therapy time per day)	10 h	10 h	10 h	8 h	8 h
CCM timing	1 h stimulation/1.23 h pause	1 h stimulation/1.23 h pause	1 h stimulation/1.23 h pause	1 h stimulation/1.59 h pause	1 h stimulation/1.59 h pause
CCM pacing voltage	7.5 V	7.5 V	6 V	7.5 V	5 V
Maximum rate for CCM inhibition	110 b/min	98 b/min	98 b/min	110 b/min	98 b/min

CCM: cardiac contractility modulation; ICD: implanted cardioverter-defibrillator; CRT: cardiac resynchronization therapy; h: hour; b/min: beats/minute; V: volts.

**Table 3 jcm-08-00588-t003:** LVEF, NYHA class, and 6MWT at baseline and last follow-up.

	Patient 1	Patient 2	Patient 3	Patient 4	Patient 5
**LVEF (%) at implantation**	20	22	20	35	30
**LVEF (%) at 6-month follow-up**	26	34	NA	37	36
**NYHA class at implantation**	III	IV	IV	III	III
**NYHA class at 6-month follow-up**	II	II	NA	II	II
**6MWT at implantation**	290	350	210	380	320
**6MWT at 6-month follow-up**	467	460	NA	497	440

LVEF: left ventricle ejection fraction; NYHA: New York Heart Association; 6MWT: 6-min walk test.

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
