# Peer review of "New Generation Cardiac Contractility Modulation Device—Filling the Gap in Heart Failure Treatment"

_jcm, 2019, doi:10.3390/jcm8050588_

Reviewer 1 Report

The abstract does not give the immediate picture of how the study was conducted and its main results. Hypothesis and aims are poorly described and significant background literature is missing in the introduction. Data quality is poor. A proper statistical analysis is missing. Some of the inferences are largely speculative.

Author Response

Dear Reviewer,

Thank you for helping us to improve our paper.

This was not a study, but a small case series report concerning the use of a latest generation CCM devices in patients with HFrEF. This paper was not intended to be a literature review - we added some background literature as indicated.

Regarding statistical analysis, we presented only five patients, therefore statistical data analysis is not appropriate. If the sample size it too small, it will not yield valid results. Moreover, the results from the small sample size will be questionable.

Yours sincerely,

Professor Diana Tint

Reviewer 2 Report

I had the pleasure to review “New generation cardiac contractility modulation device – filling the gap in heart failure treatment”. Diana Tint et al describe a series of five patients receiving cardiac contractility modulation (CCM) devices. In the main text, the indication, implantation and follow-up are described very carefully. However, there are major flaws that do not allow a publication of this version of the manuscript.

First, the authors’ data and their conclusion do not match. In the text, they describe that one patient died a few weeks after the implantation (maybe as consequence of the procedure?) and in other patients, life-saving functions of the implanted cardioverter defibrillator had to be deactivated to avoid cross-talk. In contrast, they describe CCM to be safe and effective in all patients with severe heart failure. In the abstract, the authors write that CCM should be limited to patients with severe heart failure without giving any evidence for this notion. Furthermore, LVEF is described to ameliorate during follow-up, but the authors did not perform any statistical test to prove their hypothesis.

Second, there are some other inconsistencies that lead to misunderstanding of some areas of the manuscript. For instance, in the abstract it is noted that all patients reach NYHA class II although from the abstract it is not known where they started. Furthermore, the title is misleading as in my opinion it would refer to a review.

There are also minor flaws concerning the formatting. Affiliation #3 is not used, numbers are written with either no, one or two decimals, row “digoxin” is empty in Table 1 and the authors do not report if conflicts of interest exist.

I would recommend the following further adjustments:

-        Please delete column 2 (CCM device) in Table 2 and switch columns and rows in this table to be consistent with other tables.

-        Lines 108-109: Please rephrase this sentence as it does not comply with the previous heading.

-        Lines 117-118: Did you have to deactivate ATPs/shocks in Patient 4 as well?

-        Lines 128-129 and 144-145: Was this the same patient?

-        Please provide a reference for the first sentence of the discussion.

Author Response

Dear Reviewer,

Thank you very much for helping us to improve our paper.

The patient who died was a very severe patient two years ago, he was placed on the waiting list for cardiac transplantation. He had bi-ventricular heart failure, thus the LBi-VAD was not an option for him. The CCM therapy remained the only potential option for this relatively young patient in order to prolong his life. His outcome was unfavorable, but most probably due to his very severe status and lack of myocardial resources.

No lifesaving (ATP/shock function) was inactivated in any patient. In order to make those things clearer, we specifically add a comment in the body text.

In this patient the RV lead was deactivated for pacing, not for delivering ATP and/or Shock. The LV pacing was kept (pacing in fusion), thus the contraction synchrony was maintained.

In P2 we deactivated one of the two the RV leads from CCM device, not from the ICD. Thus, the ICD therapy was not affected.

The initial and final status of NYHA class was depicted in Table 3. At the time of the implant 3 patients (P1, P4 and P5 were in NYHA III class, while P2 and P3 were in class IV NYHA). We added a sentence in the text regarding this aspect.

The title suggest that CCM may offer a therapy for those patients who are not CRT candidates or who are CRT non-responders, and/or have atrial fibrillation. All those conditions are referred as a “therapeutic gap” which may be at least partially filled with the aid of CCM therapy provided by the new CCM generation device. If the reviewer has other suggestion, we are happy to take it into consideration.

We agree with the reviewer regarding the conclusion inconsistency. We modified the conclusion accordingly and inserted a comment at the end of the discussion section.

Regarding statistical analysis, we presented only five patients (a case series). The number of patients is not suitable for an appropriate statistical analysis. If the sample size it too small, it will not yield valid results.  Moreover, the results from the small sample size will be questionable.

We modified Table 2 as suggested.

Line 128-129 and 144 and 145 – yes, it was the same patient. In first paragraph the comments were related to technical issues, while in the second one we focused on the outcome (re-hospitalizations, deaths, etc).

The digoxin row was filled with data.

We changed the affiliations.

We provided references for the first sentence of the discussion section.

The conflict of interests declaration was provided.

Yours sincerely,

Professor Diana Tint

Reviewer 3 Report

General Comments:
1. Tint et al. submitted a well-written and structured small case series report on the use of latest generation CCM devices in HFrEF patients with varying severity. The results are promising and in accordance with the pertinent literature on this subject matter.
2. With the exception of patient 3 (whose unfortunate and untimely demise arguably further suggests that he should not have been included in the first place since such an outcome was probably to be expected), the cases reported by Tint et al. provide further evidence of substantial benefits of CCM therapy for a large group of HFrEF patients, even with LVEF<30%.
3. Since the submitted manuscript is rather short, in this reviewer's opinion it might be a good idea to provide some additional background/discussion of particularly relevant topic, e.g. (but not limited to) duration of CCM delivery (how many hours per day), a short discussion of the mechanism of action of CCM, and a greater description of differences of the newest device compared to predecessors and the implications this has for therapy.

Author Response

Reviewer 3

Top of Form

Dear Reviewer,

Thank you very much for helping us to improve our article.

We added a further paragraph in introduction section concerning the mechanisms by which this therapy improves cardiac contractility, along with some additional technical details related to programming options of the impulse voltage, duration and length of the therapy deliverance.

Yours sincerely

Professor Diana Tint

Round  2

Reviewer 1 Report

The Authors did not adequately address the concerns raised by this reviewer.

Major flaws remain.

Author Response

Dear Reviewer,

We tried to change our article according to your suggestions, but some of your queries cannot be addressed.

As previously explained, this is not a clinical study, and so it does not have a specific design: it is a case series of 5 patients, most of the data was retrospectively collected and, just like in all papers presenting case series, statistical analysis was not performed because it is not appropriate. The follow-up was performed according with the protocol approved in our clinic.

We presented five patients with HF, with different clinical presentation, and we shared the impact of CCM therapy delivery on these patients. Some important and useful technical aspects regarding device-to-device interaction were depicted. Our results are in line with the existing scientific literature reports and provide further evidence of the benefits of CCM therapy.

It has to be emphasized, however, that the CCM device therapy is relatively new, and there are only some case reports or case series reports already published in prestigious journals; there is also a multi-centric prospective study comprising only 17 patients and moreover, the most important randomized trial gathered only 74 cases with CCM. Because of these aspects, we believe that our clinical experience worth sharing.

We hope you will reconsider your position regarding this paper.

Yours sincerely,

Professor Diana Tint

Reviewer 2 Report

The authors made substantial improvements to the manuscript. However, there are still some minor issues that should be considered:

Please adjust numbers to have the same number of decimals. For example, I would recommend to write 67±8 instead of 67±8.03 years, 310±65 instead of 310±65.19 m, 25.4±6.8 instead of 25.4±6.76%. I would suggest to adjust the number of decimals in the abstract, at lines 120-121  and 159.

Please define NYHA at the first instance (at line 60 instead of line 66).

There is some inconsistency about the pacing voltage and pacing duration of the CCM system. In line 43, you write that the voltage can be set between 4.0 and 7.5V, while at lines 90 and 102 you write values of 5-7.5 V. According to line 44, the pulse duration can be set to 5.14 to 6.60 msec, while at line 90 the total train duration is considered 20.5-22.5msec.

You write the same sentence ("the system is designed to modulate the strength...") twice at lines 45-46 and 57-58.

At line 163, authors should state if the unfavourable outcome of P3 was linked to the procedure or not. Thank you for including this patient into the case series, as it highlights the severity of HF and comorbidities of patients receiving CCM therapy.

Author Response

Dear Reviewer,

Thank you for helping us to further improve our article.

We adjusted the numbers to one decimal format, according to indications.

We defined NYHA class at first instance.

The different numbers and intervals referred to voltage values and phase duration appear because these are programmable parameters. Thus, the voltage could be set according to the patient’s tolerance with a minimum value of 4 V and a maximum 7.5 V (this is the 4-7.5V interval), targeting the maximum tolerate value. In our case series, we had a patient in whom we had to program a lower stimulation value (5 V), while all others tolerated the maximum of 7.5 V (thus the interval was 5-7.5 V).

The phase duration of the impulse is also a programmable parameter, and it is automatically set by the device. Each stimulus train consists of two biphasic impulses, thus the total train duration is generally 20.5-22.5 msec. We added an explanatory sentence in the text in order to make this information clearer.

The sentence "the system is designed to modulate the strength” was rephrased in lines 45-46.

We stated that the unfavorable outcome in P3 was not related to the procedure.

The paper was corrected for English spelling and grammar.

Yours sincerely,

Professor Diana Tint